# Direct Transport to Cardiac Arrest Center and Survival Outcomes after Out-of-Hospital Cardiac Arrest by Urbanization Level

**DOI:** 10.3390/jcm11041033

**Published:** 2022-02-16

**Authors:** Eujene Jung, Young Sun Ro, Jeong Ho Park, Hyun Ho Ryu, Sang Do Shin

**Affiliations:** 1Department of Emergency Medicine, Chonnam National University Hospital, Gwangju 61469, Korea; 81823ej@hanmail.net (E.J.); oriryu@hanmail.net (H.H.R.); 2Department of Emergency Medicine, Chonnam National University Medical School, Gwangju 61469, Korea; 3Department of Emergency Medicine, Seoul National University Hospital, Seoul 03080, Korea; timthe@gmail.com (J.H.P.); sdshin@snu.ac.kr (S.D.S.); 4Department of Emergency Medicine, Seoul National University College of Medicine, Seoul 03080, Korea; 5Laboratory of Emergency Medical Services, Seoul National University Hospital Biomedical Research Institute, Seoul 03080, Korea

**Keywords:** out-of-hospital cardiac arrest, post-resuscitation care, outcomes

## Abstract

Current guidelines for post-resuscitation care recommend regionalized care at a cardiac arrest center (CAC). Our objectives were to evaluate the effect of direct transport to a CAC on survival outcomes of out-of-hospital cardiac arrests (OHCAs), and to assess interaction effects between CAC and urbanization levels. Adult EMS-treated OHCAs with presumed cardiac etiology between 2015 and 2019 were enrolled. The main exposure was the hospital where OHCA patients were transported by EMS (CAC or non-CAC). The outcomes were good neurological recovery and survival to discharge. Multivariable logistic regression analyses were conducted. Interaction analysis between the urbanization level of the location of arrest (metropolitan or urban/rural area) and the exposure variable was performed. Among the 95,931 study population, 23,292 (24.3%) OHCA patients were transported directly to CACs. Patients in the CAC group had significantly higher likelihood of good neurological recovery and survival to discharge than the non-CAC group (both *p* < 0.01, aORs (95% CIs): 1.75 (1.63–1.89) and 1.70 (1.60–1.80), respectively). There were interaction effects between CAC and the urbanization level for good neurological recovery and survival to discharge. Direct transport to CAC was associated with significantly better clinical outcomes compared to non-CAC, and the findings were strengthened in OHCAs occurring in nonmetropolitan areas.

## 1. Introduction

Out-of-hospital cardiac arrest (OHCA) is a major global health problem, with high incidence and poor survival outcomes [1,2]. Despite extensive efforts to increase resuscitation and post-resuscitation care, mortality and disability rates remain high, with only 7–10% of OHCA patients surviving to discharge and only less than 5% of those patients discharging with favorable neurological recovery [2,3]. The emergency medical services (EMS) personnel are responsible for on-scene and during-transport resuscitation and transporting of OHCA patients to the appropriate hospital for post-resuscitation care [4].

Regional systems of care involving centralization of post-resuscitation care have been proposed to improve survival outcomes of OHCA, as OHCA is considered to be best treated in regional hospitals with highly resource-intensive treatments such as extracorporeal membrane oxygenation, percutaneous cardiac intervention (PCI), and targeted temperature management (TTM) [5,6]. Current guidelines for post-resuscitation care recommend regionalization to designated cardiac arrest centers (CAC) that can provide 24 h immediate PCI and can provide TTM [5,7]. However, in recent systematic review and meta-analysis studies, it was suggested that while transporting patients directly to a CAC did associate with improved clinical outcomes at hospital discharge, it did not improve 1-month survival outcomes [8,9]. Sudden cardiac arrest is one of the most time-sensitive diseases, and the increased transport time interval in bypassing the nearest hospital to reach the destination hospital may be detrimental for some OHCA patients with specific conditions [10]. 

It is hypothesized that the direct transport to a CAC would improve overall survival outcomes in OHCA patients, while that effect will vary depending on the urbanization level. The urbanization level affects the distribution of CAC as a surrogate indicator of community, EMS, and resources of hospital resuscitation, and is one of the potential risk factors for survival outcomes of OHCA [11]. The objectives of this study were to evaluate effects of the direct transport to a CAC on clinical outcomes of OHCA patients with presumed cardiac etiology, and to assess whether the effects vary across the urbanization level of the location where OHCA occurred.

## 2. Materials and Methods

### 2.1. Study Design and Setting

This is a cross-sectional study, using a nationwide, population-based prospective registry of OHCAs including all patients transported by EMS in Korea.

Korea has approximately 50 million people living in 100,210 km^2^, and there are 17 provinces. These areas are subdivided into 229 counties for administrative purposes, including 69 counties in metropolitan cities (median population density: 9214 persons per km^2^), 78 counties in urban cities (median population density: 598 persons per km^2^), and 82 counties in rural areas (median population density: 65 persons per km^2^) [12]. Of the 17 provinces, 7 provinces are metropolitan cities, and 10 provinces have a mix of urban cities and rural areas.

The EMS system of Korea is a government-based system operated by 17 provincial headquarters of the National Fire Agency. The EMS personnel perform basic life support on scene and during transport with advanced airway management and intravenous fluid administration. Since declaration of death in the field is not permitted for EMS providers, all EMS-treated OHCA victims are transported to the nearest emergency department (ED) based on the standard operation protocol.

In Korea, there are 402 EDs that are categorized into three levels by the government according to capacity and resources such as equipment, staffing, and size of the ED: level-1 EDs (*n* = 38), level-2 EDs (*n* = 128), and level-3 EDs (*n* = 236). All EDs generally perform acute cardiac management and post-resuscitation care in accordance with international standard guidelines such as the 2020 American Heart Association guidelines [13]. Most of the level-1 and level-2 EDs perform post-resuscitation care such as PCI and TTM. The Ministry of Health and Welfare has designated and operated cardiovascular disease regional centers but has not yet designated and/or certified cardiac arrest-specific regional centers.

### 2.2. Data Sources

This study identified a study population using the Korean nationwide OHCA registry, which captures all EMS-assessed OHCA patients across the country. The OHCA registry is a prospective observational registry that was launched in 2006 through a collaboration between the National Fire Agency and the Korean centers for Disease Control and Prevention (CDC). The registry includes ambulance run sheets, dispatch records, EMS cardiac arrest in-depth registry, and medical record reviews. The medical record reviewers from Korea CDC extract data regarding the etiology, hospital care, and outcomes based on the Utstein guidelines. The project quality management committee (QMC) is composed of emergency medicine physicians, cardiologists, epidemiologists, statistical experts, and medical record review experts. All of the items, including definitions, inclusion and exclusion criteria, examples, and warnings, are defined in the medical record review guidelines and were developed by the project QMC. Explanations on nationwide OHCA registry, detailed data collection process, and quality management protocols are reported in previous studies [14]. 

### 2.3. Study Population

This study included all EMS-treated OHCA patients aged 18 or over with presumed cardiac etiology between January 2015 and December 2019. Patients whose arrest occurred in an ambulance during transport or witnessed by EMS personnel were excluded. The cause of arrest was presumed to be cardiac if there was no evident noncardiac cause such as asphyxia, drowning, trauma, poisoning, and burn. The cause of arrest was measured by medical record reviewers with discharge summary abstracts or medical records written by the inpatient care doctors, based on the Utstein guidelines [15].

### 2.4. Main Outcomes

The primary and secondary outcomes were good neurological recovery and survival to discharge. Good neurological recovery was defined as cerebral performance category I (good cerebral performance; no neurologic disability) or II (moderate cerebral disability; able to perform daily activities independently) at time of hospital discharge.

### 2.5. Measurements and Variables

The main exposure of this study is the type of hospital that patients are transported to by EMS providers, which is classified as either CAC or non-CAC. In this study, CAC was defined as a hospital that performed both PCI and TTM at least once each year during the study period for OHCA patients [5].

Patient arrest information, including age, sex, comorbidities (diabetes mellitus, hypertension, and heart disease), urbanization level of arrest location (metropolitan or urban/rural area), and place of arrest (public or private), were collected. Prehospital EMS information including witness status, bystander CPR, initial electrocardiogram rhythm (shockable or non-shockable), EMS time variables (response time interval (time from the call to ambulance arrival at the scene), scene time interval (time from ambulance arrival to departure from the scene), and transport time interval (time from departure to hospital arrival)), multitier response, prehospital airway management, and mechanical CPR were retrieved. Information concerning hospital outcome-related variables, including post-resuscitation care and clinical outcomes, were also collected.

### 2.6. Statistical Analysis

A descriptive analysis was conducted to compare the characteristics of patients transported to CAC and non-CAC. Categorical variables were shown by counts and proportion and tested by chi-square test. Continuous variables were shown by medians and quartiles and tested by Wilcoxon rank-sum test, since EMS time variables have a nonparametric distribution.

Both univariable and multivariable logistic regression analysis were performed to estimate the effect of direct transport to CACs on study outcomes. Crude and adjusted odds ratios (aORs) with 95% confidence intervals (CIs) were calculated. Finally, the interaction model between the transported hospital and urbanization level of arrest location (metropolitan area or urban/rural area) was conducted to estimate whether the effects of direct transport to CACs varies across urbanization levels that affect the distribution of CAC.

A sensitivity analysis was conducted for pulseless OHCA patients who did not achieve prehospital return of spontaneous circulation (ROSC), to examine whether the association between direct transport to CAGs and study outcomes were maintained.

All statistical analysis was performed using SAS version 9.4 (SAS Institute Inc., Cary, NC, USA). *p* < 0.05 was considered statistically significant.

### 2.7. Ethics Statements

This study complies with the Declaration of Helsinki. This study was approved by the Institutional Review Board (IRB) of Seoul National University Hospital and the requirement for informed consent was waived due to the retrospective nature of this study (IRB No. SNUH-1103-153-357).

## 3. Results

Among 139,212 EMS-treated OHCA cases that occurred within the study period, 95,931 (68.9%) met the inclusion criteria. We excluded patients who were younger than 18 years old (*n* = 3124), who had noncardiac etiology (*n* = 33,004), and those whose arrest occurred during transport (*n* = 7153) (Figure 1).

### 3.1. Demographic Findings

The demographics of the study population according to the transported hospitals are presented in Table 1. Of the 95,931 eligible patients, 23,292 (24.3%) and 72,639 (75.7%) OHCA patients were transported to CAC and non-CAC hospitals, respectively. The good neurological recovery and survival to discharge rates were 9.1% and 13.4% in the CAC group, and 4.7% and 7.4% in the non-CAC group, respectively (both *p*-value < 0.01). OHCAs in the CAC group occurred more frequently in metropolitan areas compared to the non-CAC group (67.1% vs. 32.1%, *p* < 0.01), and hospital treatments including TTM, PCI, and ECMO were also performed more frequently in the CAC group. The medians (interquartile ranges) of EMS transport time interval from scene to hospital were 7 (5–10) min in the CAC group and 6 (4–11) min in the non-CAC group (*p* < 0.01).

The demographics of OHCA patients according to the urbanization level of arrest location are summarized in Table 2. Of the eligible patients, 38,939 (40.6%) cases of OHCA occurred in metropolitan areas and 56,992 (59.4%) cases occurred in nonmetropolitan areas. OHCA cases occurring in metropolitan areas were transported more frequently to a CAC (40.1% vs. 13.5%, *p* < 0.01). Good neurological recovery and survival to discharge rates were 7.0% and 10.7% in the metropolitan group, and 5.0% and 7.6% in the urban/rural group, respectively (both *p*-value < 0.01).

### 3.2. Main Results

The results of multivariable logistic regression analyses are shown in Table 3. After adjustments for potential confounders, patients who were transported to CAC hospitals had significantly higher likelihood of good neurological recovery at hospital discharge and survival to discharge than those transported to non-CAC hospitals (aORs (95% CIs): 1.75 (1.63–1.89) and 1.70 (1.60–1.80), respectively).

### 3.3. Interaction Analysis

In the interaction analysis, statistically significant interaction effects were found between transported hospital and urbanization level (Table 4). The aORs for the outcomes of the CAC group and non-CAC differed depending on the urbanization level of area of OHCA. The CAC hospitals had interaction effects for good neurological recovery at discharge (aORs (95% CIs): 1.51 (1.40–1.63) for patients in metropolitan areas vs. 1.98 (1.81–2.17) for patients in urban/rural areas, and survival to discharge (aORs (95% CIs): 1.63 (1.48–1.80) for patients in metropolitan areas vs. 1.91 (1.71–2.14) for patients in urban/rural areas (both p for interaction <0.01).

### 3.4. Sensitivity Analysis

The multivariable logistic regression analysis and interaction analysis were performed for pulseless OHCA patients who did not achieve prehospital ROSC (Table 5 and Appendix A). Pulseless OHCA patients who were transported to CAC hospitals had significantly higher likelihoods of good neurological recovery and survival to discharge (aORs (95% CIs): 1.49 (1.18–1.87) and 1.45 (1.30–1.61), respectively). In the interaction analysis, the CAC hospitals had interaction effects for good neurological recovery at discharge (aORs (95% CIs): 1.36 (1.04–1.77) for patients in metropolitan areas vs. 1.84 (1.22–2.79) for patients in urban/rural areas, and survival to discharge (aORs (95% CIs): 1.24 (1.09–1.42) for patients in metropolitan areas vs. 1.91 (1.60–2.27) for patients in urban/rural areas (both *p* for interaction < 0.01).

## 4. Discussion

Using the Korean national OHCA database, this study discovered that adult OHCA patients with presumed cardiac etiology who were transported to CAC hospitals were more likely to have better survival outcomes compared to patients transported to non-CAC hospitals. In the interaction analysis, OHCAs occurring in urban/rural areas have better clinical outcomes from direct transport to a CAC hospital. These trends were maintained in the sensitivity analysis of pulseless OHCA patients who did not achieve prehospital ROSC. This research contributes to understanding the relationship between regionalization of post-resuscitation care and overall survival outcomes and will help develop strategies to improve survival outcomes in OHCAs that occur in urban/rural areas.

Regionalized systems of post-resuscitation care have been proposed to improve survival outcomes of OHCAs through centralization of highly resource-intensive treatments such as TTM, acute cardiac care including PCI, and multimodal neuro-intensive care. In recent systematic review and meta-analysis, direct transport of OHCA patients to CACs by EMS providers was associated with increased survival outcomes, despite very low certainty of evidence [8,9]. In meta-analysis studies, the definition of CAC varied from study to study, and the capability of PCI was essential, while availability for TTM was treated as important. Although the definition of CAC is used in various ways in different countries based on international guidelines [16], direct transport to a PCI-capable hospital increased overall survival and neurological outcomes of OHCAs [17,18]. Only 11% of OHCA patients who were transported to CAC hospitals had received PCI in this study. Even so, the good clinical outcomes of patients in the CAC group had probably been impacted by capability of PCI as well as the accumulated experience and ability of these CAC hospitals to treat OHCA patients [9]. 

However, direct transport of OHCA patients to CAC hospitals results in increased transport time interval for some patients [19]. A previous study related to prehospital transport time of OHCAs reported that delaying hospital arrival time by about 14 min counteracted the potential benefit of transporting them to a PCI-capable center [20,21]. Because current studies focusing on the relationship between distance from scene to hospital and clinical outcomes are mainly conducted in urban environments and limited to relatively short transport time intervals, there is insufficient evidence to agree to current guidelines stating to bypass the nearest hospital and transport to CAC hospitals in rural areas where transport distances may be substantially longer [22,23].

In the interaction analysis of the study, OHCAs occurring in nonmetropolitan areas had higher odds of survival outcomes with direct transport to a CAC hospital compared to OHCAs in metropolitan areas (*p*-for-interaction < 0.01). One hypothesis that may explain this result is that the clinical capabilities of non-CAC hospitals located in nonmetropolitan areas, including manpower, equipment, and facilities, are below those in metropolitan area. As the beneficial effects of regionalization of post-resuscitation care for OHCAs have been strengthened in urban/rural areas, CACs should be designated and invested in to achieve centralization of resource-intensive care to improve the survival outcomes of OHCAs in nonmetropolitan areas [24].

Characteristics of destination hospitals are associated with the clinical outcomes of OHCA, including level of EDs, urbanization level of location of hospitals, teaching status, and OHCA case volume [25,26,27,28]. In this study, direct transport of OHCA patients to a hospital where post-resuscitation care is capable showed favorable survival and neurological outcomes, and these results were reinforced in OHCA patients occurring in nonmetropolitan (urban/rural) areas. To improve the survival outcomes of OHCAs, it is considerable to designate and operate CACs to achieve a centralization of resource-intensive post-resuscitation care in nonmetropolitan areas.

This study has a number of limitations. First, since there were no designated cardiac arrest-specific regional centers in Korea, CAC was defined as a hospital that performed PCI and TTM in this study. The definition of CAC varied from study to study [8], which meant the definition would have affected the study results. The generalizability of findings of this study to other countries needs to be further evaluated. Second, the definition of urbanization level (metropolitan vs. urban/rural area) may not accurately reflect the level of medical resources because it is population-based and therefore such classification may have influenced the study results. Third, it is difficult to perform a geospatial analysis to evaluate the proportion of OHCA patients who were not transported to the nearest ED and their EMS transport time interval in this study. Fourth, this study is an observational retrospective analysis, which may have introduced some unmeasured confounders as is known with this study design. Lastly, while we used multivariable analysis, unmeasured and unmeasurable confounders may have influenced the clinical outcomes of the study.

## 5. Conclusions

Direct transport of OHCA patients to cardiac arrest centers was associated with significantly higher survival and favorable neurological outcomes compared to patients transported to non-CAC hospitals. Furthermore, the findings were consistent and strengthened in OHCAs occurring in nonmetropolitan areas. Designating and investing in CACs to achieve centralization of post-resuscitation care could improve the survival outcomes of OHCAs, especially in nonmetropolitan areas.

## Figures and Tables

**Figure 1 jcm-11-01033-f001:**
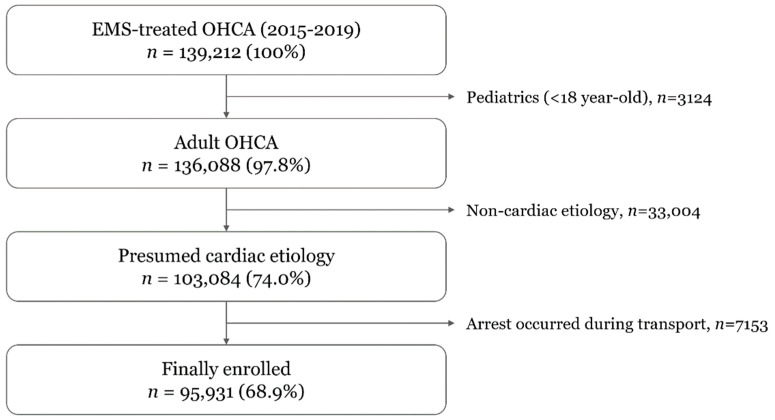
Patient flow. EMS, emergency medical services; OHCA, out-of-hospital cardiac arrest.

**Table 1 jcm-11-01033-t001:** Characteristics of out-of-hospital cardiac arrest patients according to transported hospitals.

	Total	CAC	Non-CAC	*p*-Value
	*n* (%)	*n* (%)	*n* (%)	
Total	95,931 (100.0)	23,292 (100.0)	72,639 (100.0)	
Age, year				<0.01
19–65	31,722 (33.1)	8460 (36.3)	23,262 (32.0)	
65–120	64,209 (66.9)	14,832 (63.7)	49,377 (68.0)	
Sex, female	35,089 (36.6)	8110 (34.8)	26,979 (37.1)	<0.01
Comorbidity				
Diabetes mellitus	21,386 (22.3)	5922 (25.4)	15,464 (21.3)	<0.01
Hypertension	32,708 (34.1)	8988 (38.6)	23,720 (32.7)	<0.01
Heart disease	16,577 (17.3)	4523 (19.4)	12,054 (16.6)	<0.01
Metropolitan area	38,939 (40.6)	15,619 (67.1)	23,320 (32.1)	<0.01
Place of arrest, public	19,202 (20.0)	5070 (21.8)	14,132 (19.5)	<0.01
Arrest witnessed	44,429 (46.3)	11,659 (50.1)	32,770 (45.1)	<0.01
Bystander CPR	57,803 (60.3)	13,946 (59.9)	43,857 (60.4)	0.17
Initial shockable rhythm	16,237 (16.9)	4732 (20.3)	11,505 (15.8)	<0.01
Response time interval, min				<0.01
0–3	5123 (5.3)	1287 (5.5)	3836 (5.3)	
4–7	50,555 (52.7)	13,882 (59.6)	36,673 (50.5)	
8–	40,253 (42.0)	8123 (34.9)	32,130 (44.2)	
Median (IQR)	7 (5–9)	7 (5–10)	7 (5–10)	<0.01
Scene time interval, min				<0.01
0–10	28,566 (29.8)	5761 (24.7)	22,805 (31.4)	
11–15	33,584 (35.0)	8280 (35.5)	25,304 (34.8)	
16–	33,781 (35.2)	9251 (39.7)	24,530 (33.8)	
Median (IQR)	13 (10–18)	14 (11–18)	13 (10–17)	<0.01
Transport time interval, min				<0.01
0–3	17,312 (18.0)	3184 (13.7)	14,128 (19.4)	
4–7	39,459 (41.1)	10,588 (45.5)	28,871 (39.7)	
8–	39,160 (40.8)	9520 (40.9)	29,640 (40.8)	
Median (IQR)	6 (4–10)	7 (5–10)	6 (4–11)	<0.01
Multitier response	55,631 (58.0)	16,304 (70.0)	39,327 (54.1)	<0.01
EMS management				
Advanced airway	59,978 (62.5)	16,650 (71.5)	43,328 (59.6)	<0.01
Mechanical CPR	10,866 (11.3)	4138 (17.8)	6728 (9.3)	<0.01
ED level				<0.01
Level 1	17,778 (18.5)	10,032 (43.1)	7746 (10.7)	
Level 2	45,389 (47.3)	13,116 (56.3)	32,273 (44.4)	
Level 3	32,764 (23.2)	144 (0.6)	32,630 (44.9)	
Hospital treatment				
TTM	2938 (3.1)	1874 (8.0)	1064 (1.5)	<0.01
PCI	5879 (6.1)	2521 (10.8)	3358 (4.6)	<0.01
ECMO	936 (1.0)	476 (2.0)	460 (0.6)	<0.01
Survival outcomes				
Survival to discharge	8465 (8.8)	3120 (13.4)	5345 (7.4)	<0.01
Good neurological recovery	5563 (5.8)	2113 (9.1)	3450 (4.7)	<0.01

CAC, cardiac arrest center; CPR, cardiopulmonary resuscitation; IQR, interquartile range; ED, emergency department; TTM, targeted temperature management; PCI, percutaneous coronary intervention; ECMO, extracorporeal membrane oxygenation.

**Table 2 jcm-11-01033-t002:** Characteristics of out-of-hospital cardiac arrest patients according to urbanization level of arrest location.

	Total	Metropolitan	Urban/Rural	*p*-Value
	*n* (%)	*n* (%)	*n* (%)	
Total	95,931 (100.0)	38,939 (100.0)	56,992 (100.0)	
Cardiac arrest center				<0.01
Yes	23,292 (24.3)	15,619 (40.1)	7673 (13.5)	
No	72,639 (75.7)	23,320 (59.9)	49,319 (86.5)	
Age, year				<0.01
19–65	23,292 (24.3)	15,619 (40.1)	7673 (13.5)	
65–120	72,639 (75.7)	23,320 (59.9)	49,319 (86.5)	
Sex, female	64,209 (66.9)	25,674 (65.9)	38,535 (67.6)	<0.01
Comorbidity				
Diabetes mellitus	35,089 (36.6)	13,801 (35.4)	21,288 (37.4)	<0.01
Hypertension	21,386 (22.3)	9441 (24.2)	11,945 (21.0)	<0.01
Heart disease	32,708 (34.1)	14,135 (36.3)	18,573 (32.6)	<0.01
Place of arrest, public	19,202 (20.0)	8008 (20.6)	11,194 (19.6)	<0.01
Arrest witnessed	44,429 (46.3)	18,409 (47.3)	26,020 (45.7)	<0.01
Bystander CPR	57,803 (60.3)	22,538 (57.9)	35,265 (61.9)	<0.01
Initial shockable rhythm	16,237 (16.9)	7086 (18.2)	9151 (16.1)	0.01
Response time interval, min				<0.01
0–3	5123 (5.3)	2269 (5.8)	2854 (5.0)	
4–7	50,555 (52.7)	25,074 (64.4)	25,481 (44.7)	
8–	40,253 (42.0)	11,596 (29.8)	28,657 (50.3)	
Median (IQR)	7 (5–9)	6 (5–8)	8 (6–11)	<0.01
Scene time interval, min				<0.01
0–10	28,566 (29.8)	10,677 (27.4)	17,889 (31.4)	
11–15	33,584 (35.0)	15,132 (38.9)	18,452 (32.4)	
16–	33,781 (35.2)	13,130 (33.7)	20,651 (36.2)	
Median (IQR)	13 (10–18)	13 (10–17)	13 (10–18)	<0.01
Transport time interval, min				<0.01
0–3	17,312 (18.0)	7366 (18.9)	9946 (17.5)	
4–7	39,459 (41.1)	19,765 (50.8)	19,694 (34.6)	
8–	39,160 (40.8)	11,808 (30.3)	27,352 (48.0)	
Median (IQR)	6 (4–10)	6 (4–8)	7 (4–12)	<0.01
Multitier response	55,631 (58.0)	26,993 (69.3)	28,638 (50.2)	<0.01
EMS management				
Advanced airway	59,978 (62.5)	27,384 (70.3)	32,594 (57.2)	<0.01
Mechanical CPR	10,866 (11.3)	6947 (17.8)	3919 (6.9)	<0.01
ED level				<0.01
Level 1	17,778 (18.5)	7336 (18.8)	10,442 (18.3)	
Level 2	45,389 (47.3)	22,232 (57.1)	23,157 (40.6)	
Level 3	32,764 (23.2)	9371 (24.1)	23,393 (41.1)	
Hospital treatment				
TTM	2938 (3.1)	1745 (4.5)	1193 (2.1)	<0.01
PCI	5879 (6.1)	3185 (8.2)	2694 (4.7)	<0.01
ECMO	936 (1.0)	548 (1.4)	388 (0.7)	<0.01
Survival outcomes				
Survival to discharge	8465 (8.8)	4155 (10.7)	4310 (7.6)	<0.01
Good neurological recovery	5563 (5.8)	2711 (7.0)	2852 (5.0)	<0.01

CPR, cardiopulmonary resuscitation; IQR, interquartile range; ED, emergency department; TTM, targeted temperature management; PCI, percutaneous coronary intervention; ECMO, extracorporeal membrane oxygenation.

**Table 3 jcm-11-01033-t003:** Multivariable logistic regression models for study outcomes.

	Total	Outcome		Model 1	Model 2	Model 3
	*n*	*n*	%	aOR (95% CI)	aOR (95% CI)	aOR (95% CI)
Good neurological recovery						
Transported hospital						
Noncardiac center	72,639	3450	4.7	1.00	1.00	1.00
Cardiac center	23,292	2113	9.1	1.81 (1.70–1.90)	1.66 (1.55–1.79)	1.75 (1.63–1.89)
Urbanization level						
Urban/rural area	56,992	2852	5.0	1.00	1.00	1.00
Metropolitan area	38,939	2711	7.0	1.18 (1.12–1.26)	1.17 (1.09–1.25)	1.05 (0.98–1.13)
Survival to discharge						
Transported hospital						
Noncardiac center	72,639	5345	7,4	1.00	1.00	1.00
Cardiac center	23,292	3120	13.4	1.75 (1.67–1.85)	1.62 (1.53–1.72)	1.70 (1.60–1.80)
Urbanization level						
Urban/rural area	56,992	4310	7.6	1.00	1.00	1.00
Metropolitan area	38,939	4155	10.7	1.23 (1.18–1.30)	1.23 (1.16–1.30)	1.12 (1.05–1.18)

aOR, adjusted odds ratio; CI, confidence interval. Model 1: adjusted for age and sex. Model 2: adjusted for variables in Model 1, comorbidities (diabetes mellitus, hypertension, and heart disease), place of arrest, witness status, bystander CPR, and initial shockable rhythm. Model 3: adjusted for variables in Model 2, response time interval, scene time interval, transport time interval, multitier response, EMS airway management, and mechanical CPR.

**Table 4 jcm-11-01033-t004:** Interaction analysis between direct transport to cardiac arrest centers and urbanization level.

	Transported Hospital				
	Non-CAC	Cardiac Arrest Center			*p*-for-Interaction
		aOR	95% CI		
Good neurological recovery					
Urbanization level					<0.01
Metropolitan area	ref.	1.51	1.40	1.63	
Urban/rural area	ref.	1.98	1.81	2.17	
Survival to discharge					
Urbanization level					<0.01
Metropolitan area	ref.	1.63	1.48	1.80	
Urban/rural area	ref.	1.91	1.71	2.14	

CAC, cardiac arrest center; aOR, adjusted odds ratio; CI, confidence interval.

**Table 5 jcm-11-01033-t005:** Sensitivity analysis of pulseless OHCA patients who did not achieve prehospital ROSC.

	Transported Hospital				
	Non-CAC	Cardiac Arrest Center			*p*-for-Interaction
		aOR	95% CI		
Good neurological recovery					
Urbanization level					<0.01
Metropolitan area	ref.	1.36	1.04	1.77	
Urban/rural area	ref.	1.84	1.22	2.79	
Survival to discharge					
Urbanization level					<0.01
Metropolitan area	ref.	1.24	1.09	1.42	
Urban/rural area	ref.	1.91	1.60	2.27	

CAC, cardiac arrest center; aOR, adjusted odds ratio; CI, confidence interval.

## Data Availability

The data of this study were obtained from the Korea Centers for Disease Control and Prevention, but restrictions apply to the availability of these data and so are not publicly available.

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
