# Peer review of "Direct Transport to Cardiac Arrest Center and Survival Outcomes after Out-of-Hospital Cardiac Arrest by Urbanization Level"

_jcm, 2022, doi:10.3390/jcm11041033_

Round 1

Reviewer 1 Report

Dear authors

Thank you very much for submitting your paper to the Journal of Clinical Medicine. Your study is a cross-sectional study to explored the interaction effects between direct transport to cardiac arrest center and urbanization level on survival outcomes after out-of- hospital cardiac arrest. This study is very interesting and gives more information and decision making on clinical practice of the referral system. I have some comments on your manuscript. 

Major comment

1. From Abstract: Patients in the CAC group had significantly higher likelihood of good 22 neurological recovery and survival to discharge than the non-CAC group (aORs (95% CIs): 1.75 23 (1.63–1.89) and 1.70 (1.60–1.80), respectively).

    • P-value of neurological recovery and survival to discharge should be documented in this part

2. Topic 2.4 Main outcome: cerebral performance category I 107 or II

    • The citation to reference or description about cerebral performance should be addressed

3. Statistical analysis: Continuous variables were shown by medians and quartiles 127 and tested by Wilcoxon rank-sum test.

    • What is the test of analysis for parametric and non-parametric data?

4. The theoretical data and the reason to conduct the the interaction model between the transported hospital and urbanization level of arrest location  (metropolitan area or urban/rural area) should be described in the method part.

5. Exclusion and inclusion criteria of study should be more described in the study population part

6. Dose unknown cause of cardiac arrest among study patients?

Best wishes

Reviewer  

Author Response

Reviewer #1:

Dear authors

Thank you very much for submitting your paper to the Journal of Clinical Medicine. Your study is a cross-sectional study to explored the interaction effects between direct transport to cardiac arrest center and urbanization level on survival outcomes after out-of- hospital cardiac arrest. This study is very interesting and gives more information and decision making on clinical practice of the referral system. I have some comments on your manuscript.

(ANSWER) Thank you for the review and the valuable comments. Each comment is addressed as you pointed.

Major comment

  1. From Abstract: Patients in the CAC group had significantly higher likelihood of good neurological recovery and survival to discharge than the non-CAC group (aORs (95% CIs): 1.75 (1.63–1.89) and 1.70 (1.60–1.80), respectively).

P-value of neurological recovery and survival to discharge should be documented in this part

(ANSWER) Thank you for the review. I revised the sentence accordingly.

(REVISION: Abstract)

Patients in the CAC group had significantly higher likelihood of good neurological recovery and survival to discharge than the non-CAC group (both p <0.01, aORs (95% CIs): 1.75 (1.63–1.89) and 1.70 (1.60–1.80), respectively).

  1. Topic 2.4 Main outcome: cerebral performance category I or II

The citation to reference or description about cerebral performance should be addressed

(ANSWER) Thank you for the review. I revised the sentence accordingly.

(REVISION: Methods)

Good neurological recovery was defined as cerebral performance category I (good cerebral performance; no neurologic disability) or II (moderate cerebral disability; able to perform daily activities independently) at time of hospital discharge.

  1. Statistical analysis: Continuous variables were shown by medians and quartiles and tested by Wilcoxon rank-sum test.

What is the test of analysis for parametric and non-parametric data?

(ANSWER) Thank you for the review. In this study, only the EMS time variables (response time interval, scene time interval, and transport time interval) were treated as continuous variables in Table 1 and 2, and other continuous variables such as age were converted into categorical variables (age: 18-64 and 65-120). Since the EMS time variables have the characteristic of non-parametric distribution, these continuous variables were shown by medians and quartiles and tested by Wilcoxon rank-sum test. I revised the sentence accordingly.

(REVISION: Methods)

Continuous variables were shown by medians and quartiles and tested by Wilcoxon rank-sum test, since EMS time variables have a non-parametric distribution.

  1. The theoretical data and the reason to conduct the interaction model between the transported hospital and urbanization level of arrest location (metropolitan area or urban/rural area) should be described in the method part.

(ANSWER) Thank you for the review. I explained the reason to conduct the interaction model in the Introduction section, as “The urbanization level affects the distribution of CAC as a surrogate indicator of community, EMS, and resources of hospital resuscitation, and is one of potential risk factors for survival outcomes of OHCA.[11]”. I added more explanations in the Methods section accordingly.

(REVISION: Methods)

Finally, the interaction model between the transported hospital and urbanization level of arrest location (metropolitan area or urban/rural area) was conducted to estimate whether the effects of direct transport to CACs varies across urbanization levels that affect the distribution of CAC.

  1. Exclusion and inclusion criteria of study should be more described in the study population part

(ANSWER) Thank you for the review. I added more descriptions of the study population in the Methods section accordingly.

(REVISION: Methods)

This study included all EMS-treated OHCA patients aged 18 or over with presumed cardiac etiology between January 2015 and December 2019. Patients whose arrest occurred in an ambulance during transport or witnessed by EMS personnel were excluded. The cause of arrest was presumed to be cardiac if there was no evident non-cardiac cause such as asphyxia, drowning, trauma, poisoning, and burn. The cause of arrest was measured by medical record reviewers with discharge summary abstracts or medical records written by the inpatient care doctors, based on the Utstein guidelines.[15]

  1. Dose unknown cause of cardiac arrest among study patients?

(ANSWER) Thank you for the review. In this study, if there was no evident non-cardiac cause such as asphyxia, drowning, trauma, poisoning, and burn, the cause of arrest was presumed to be cardiac etiology. I added more descriptions of the study population in the Methods section accordingly.

(REVISION: Methods)

The cause of arrest was presumed to be cardiac if there was no evident non-cardiac cause such as asphyxia, drowning, trauma, poisoning, and burn. The cause of arrest was measured by medical record reviewers with discharge summary abstracts or medical records written by the inpatient care doctors, based on the Utstein guidelines.[15]

Reviewer 2 Report

This is a good study, well-designed, implemented, analyzed and presented. There are several details issues identified in the copy of the manuscript that I have marked up, with opportunities for improvement. It would be attractive for the authors to consider these points and to address them, at least in the discussion. But this is not required. The manuscript is suitable for publication already, apart from a handful of minor typographic and syntactical errors.

Author Response

Reviewer #2:

This is a good study, well-designed, implemented, analyzed and presented. There are several details issues identified in the copy of the manuscript that I have marked up, with opportunities for improvement. It would be attractive for the authors to consider these points and to address them, at least in the discussion. But this is not required. The manuscript is suitable for publication already, apart from a handful of minor typographic and syntactical errors.

(ANSWER) Thank you for the review and the valuable comments. Each comment is addressed as you pointed.

The definition of CAC is post-hoc and appears somewhat arbitrary. There is no prospective definition of CAC in the EMS system that might influence their decision of where to transport. On the other hand, there is a defined hierarchy of care capabilities: Levels 1,2 and 3. It would be more objective to use these levels as the independent variable of interest. At least the functional differences between these levels warrants description. Notably, there were no patients treated by Level 3 EDs in this large study. An explanation would be welcome.

The choice of an alternative definition should be justified in relation to these established categories. The authors have done a nice job under "Sensitivity analysis". That would be a good place to include an analysis of sensitivity to the definition of CAC, in contrast to these established categories. Another way to address this issue would be to include ED level in the at least one model in the multivariate analysis.

(ANSWER) Thank you for the review.

In Korea, there is no prospective definition of CAC. It is the standard operation protocol to transport all EMS-treated OHCA victims to the nearest ED. However, it is difficult to perform a geospatial analysis to evaluate the proportion of OHCA patients who were not transported to the nearest ED in this study. We performed the sensitivity analysis for pulseless OHCA patients who did not achieve prehospital ROSC. Because patients in this group were transported while performing CPR, EMS personnel most likely transported them to the nearest ED.

Because there is a possibility of confusion in Table 1 and 2, I added the variable of Level 3 ED in Table 1 and 2, (18.5% of study population treated in the Level 1 ED, 47.3% in the Level 2 ED, and 34.2% in the Level 3 ED, respectively). There were no changes in the values of Level 1 and 2 EDs.

Somewhat burying the lead. The direct finding that CAC centers improve survival should be the main point. It is a finding worthy of publication by itself. The interaction effect is interesting, but starting the title with it distracts from this significant main effect.

(ANSWER) Thank you for the review. I have changed the Title of this study accordingly.

(REVISION: Title)

Direct transport to cardiac arrest center and survival outcomes after out-of-hospital cardiac arrest by urbanization level

In this study of post-hoc defined CAC centers accessed by a national EMS system, the question of who determines what an "appropriate" hospital, with what discretion, and on what basis, is an important issue. It should be discussed more fully.

(ANSWER) Thank you for the review. I added more explanations in the Methods section accordingly.

(REVISION: Methods)

Since declaration of death in the field is not permitted for EMS providers, all EMS-treated OHCA victims are transported to the nearest emergency department (ED) based on the standard operation protocol.

This is an important issue. It would be valuable if this richly detailed data set would support analysis, by urbanization level, of how often the nearest hospital was avoided and a further hospital chosen.

(ANSWER) Thank you for the review. In Korea, it is the standard operation protocol to transport all EMS-treated OHCA victims to the nearest ED. With our study data, it is difficult to perform a geospatial analysis to evaluate the proportion of OHCA patients who were not transported to the nearest ED. I added this limitation in the Discussion section accordingly.

(REVISION: Discussion)

Third, it is difficult to perform a geospatial analysis to evaluate the proportion of OHCA patients who were not transported to the nearest ED and their EMS transport time interval in this study.

There are 3 levels of urbanization defined here, but in the analysis the 2nd and 3rd levels are lumped. Analyzing by all 3, rather than with lumping of two together, would help us understand whether there is a graded response, and would lend credibility to the study. Perhaps the data for one of the levels is too sparse to support separate analysis, though there is a lot of data here. At least, the lumping should be explained.

(ANSWER) Thank you for the review. Of the 17 provinces in Korea, 7 provinces are metropolitan cities and the 10 provinces have a mix of urban and rural areas. Therefore, urban and rural EMS environments have similar aspects. I added more explanations in the Methods section accordingly.

(REVISION: Methods)

Of the 17 provinces, 7 provinces are metropolitan cities and 10 provinces have a mix of urban cities and rural areas.

The definition of CAC is post-hoc and essential. Be careful to be clear. Does this definition mean that a CAC must have performed at least one PCI and at least one TTM per year (i.e. at least one of each)? Is this classification over the entire history of the data (4 years), or evaluated for each year?

(ANSWER) Thank you for the review. I revised the sentence accordingly.

(REVISION: Methods)

In this study, CAC was defined as a hospital that performed both PCI and TTM at least once each year during the study period for OHCA patients.[5]

This is an overstatement of the lack of finding an effect of urbanization level on neurological outcome. The failure to find this effect is only evident in Model 3, where response, scene and transport times are included in the model. This warrants mention. It also suggests a possibly illuminating additional analysis, wherein the transport time, which is key to the putative detriment of favoring a CAC over a more local hospital, is isolated.

(ANSWER) Thank you for the review. I agree with your comments, and the objectives of this study were to evaluate effects of the direct transport to a CAC on clinical outcomes of OHCA patients with presumed cardiac etiology, and to assess whether the effects varies across the urbanization level of the location where OHCA occurred. The change of adjusted ORs of urbanization level on study outcome was not the main purpose of this study, therefore I deleted the sentence accordingly.

This suggests another potentially interesting analysis, using average annual OHCA cases treated (per hospital) as an independant variable.

(ANSWER) Thank you for the review. We are conducting another studies to evaluate the associations between annual treated OHCA case volume (total OHCA cases and post-resuscitation care such as ECMO, TTM, and PCI) and clinical outcomes after OHCA.

This is a key point. It would be interesting to know how often diversion past a more local hospital occurs in this system.

(ANSWER) Thank you for the review. I added this limitation in the Discussion section accordingly.

(REVISION: Discussion)

Third, it is difficult to perform a geospatial analysis to evaluate the proportion of OHCA patients who were not transported to the nearest ED and their EMS transport time interval in this study.

See note above. This statement is only true for Model 3, where EMS time intervals are included. This should be stated more clearly.

(ANSWER) Thank you for the review. I deleted the sentence accordingly.

This is a strong assertion with substantial potential economic consequences. It is stronger than the evidence, considering the limitations, particularly post-hoc definition of CAC. It should be tempered somewhat.

(ANSWER) Thank you for the review. I changed the sentence accordingly.

(REVISION: Discussion)

To improve the survival outcomes of OHCAs, it is considerable to designate and operate CACs to achieve a centralization of resource-intensive post-resuscitation care in non-metropolitan areas.
